# Consistency or Hypocrisy? The Impact of Internal Corporate Social Responsibility on Employee Behavior: A Moderated Mediation Model

**Yi-Ping Chang** [1,2], **Hsiu-Hua Hu** [3,*] **and Chih-Ming Lin** [4]

1. Department of Business Administration, Ming Chuan University, Taipei 111, Taiwan; ypchangtyvh@gmail.com
2. Division of Nephrology, Taoyuan Branch of Taipei Veterans General Hospital, Taoyuan 330, Taiwan
3. Department of International Business, Ming Chuan University, Taipei 111, Taiwan
4. Department of Healthcare Information and Management, Ming Chuan University, Taoyuan 333, Taiwan; cmlin@mail.mcu.edu.tw
* Correspondence: shhu@mail.mcu.edu.tw

**Abstract:** Adopting social identity theory, this study examined the process linking the relations between internal corporate social responsibility (InCSR), work engagement, and turnover intention by focusing on the mediating influence of organizational identification and the moderating role of perceived corporate hypocrisy. Data were obtained from 311 medical staff (excluding supervisors and managers) of a public regional teaching hospital in Taiwan. The results revealed that employees are more dedicated to work and less inclined to leave the firm if they perceive that InCSR is implemented within the firm. However, if an employee perceives corporate hypocrisy of inconsistency between communication and actual actions, it may have the opposite effect on employees. Likewise, the higher the level of perceived corporate hypocrisy, the lesser the positive effect of InCSR on employee behavior. Finally, the implications, limitations, and suggestions for future research were discussed.

**Keywords:** perceived internal corporate social responsibility; perceived corporate hypocrisy; organizational identification; work engagement; turnover intention

## 1. Introduction

Companies actively implement corporate social responsibility (CSR) for sustainable development, which has become an increasingly important issue in the current business world. The scope of studies on corporate social responsibility (CSR) has conventionally been limited to the relations between organizational practices and their consequences, such as a firm's financial performance and consumer behavior in general [1–3]. Nonetheless, stakeholders are the most important components for a firm in successfully implementing CSR activities and scholars believe that employees are the major essential stakeholders [4,5]. In light of this, recent studies focused on the individual level to analyze employees in psychological, attitude, and behavioral responses after perceiving CSR initiatives [6–8].

Social identity theory (SIT) is the most used basic mechanism to explain and predict an individual's response to corporate social responsibility [6,9]. Research evidence showed that many of the various consequences of CSR activities related to employees will have positive effects, such as organizational attractiveness to potential employees [10], organizational commitment [11], organizational citizenship behavior [12], job satisfaction [13,14], and employee loyalty [15]. From previous empirical research, it was also found that organizational identification plays a key role in mediating the influence of certain corporate social responsibility initiatives [6]. However, as scholars pay more attention to the positive outcomes of CSR activities, some researchers have raised certain concerns. Specifically, when stakeholders discover from other sources of information that the firm's CSR statement is inconsistent with its actual actions, this creates a perception of corporate hypocrisy

and it may have a significant impact on the firm [16]. As employees perceive that the organization's actual actions are inconsistent with its words or communication, will it affect the effectiveness of internal CSR activities? Previous studies rarely discuss the impact of employee behaviors in the awareness of perceiving corporate inconsistencies and there are also few studies in the literatures that focused on its related outcomes [17–19].

This study not only discusses the mediating role of organizational identification between employees' perception of the firm's internal CSR activities and their behavior outcomes such as work engagement and turnover intention, but also explores whether perceived corporate hypocrisy as a moderator and the degree of influence if employees sense that the actions of the firms they work for are inconsistent with their internal CSR plan. We expect this research to fill the gap in existing studies.

This study makes three contributions to existing literature. First, it puts forth a new perception related to internal CSR that corporate hypocrisy perceived by employees is a negative presentation of internal CSR practices. According to existing literature on corporate hypocrisy, most discussions focus on the negative impact on consumers and shareholders [16,20–22]. However, from the employees' perspective, what is the impact created if they sense that the firm's actual actions are "inconsistent with its promises"? Research methods based on the micro-level and the moderating effect of employee perception of corporate hypocrisy of inconsistency between communication and actual actions on work behaviors related to the firm's internal CSR activities is discussed. The research findings provide theoretical and practical evidence.

The second contribution of this study is its verification of the impact of internal CSR activities on employees' work engagement and turnover intention based on the social identity theory with organizational identification as the mediator. This represents definite progress given that previous studies mainly use the social exchange theory as the basis for explaining the relationship between CSR activities and employee behavior [23–25], even in light of some scholars pointing out the increasing significance of the identity-based driver in employee behavioral studies [26,27].

Finally, when a firm acts differently as opposed to its CSR commitment to staff, the result may lead to a negative attitude and behavior at work. It is important to understand which psychological mechanisms might have such an effect on employees [28].This study takes the perception of corporate hypocrisy as the moderating variable, and explores whether this degree of hypocritical perception of inconsistency between communication and actual actions will affect the mediating effect of organizational identification and whether it will offset the positive results caused by organizational identification, thereafter influencing employees' subsequent behavior and attitude. This paper is structured as follows. First, an introduction about the research purpose is presented. In Section 2, we describe the literature review and the development process of the theoretical framework and hypothesis. In Section 3, we explain the research method and statistical analysis. Section 4 presents research results and in Section 5, we present discussion. Finally, Section 6 contains the main conclusion and implication. Figure 1 depicts the research framework of this study. We expect this study to provide theoretical and practical directions for firm managers.

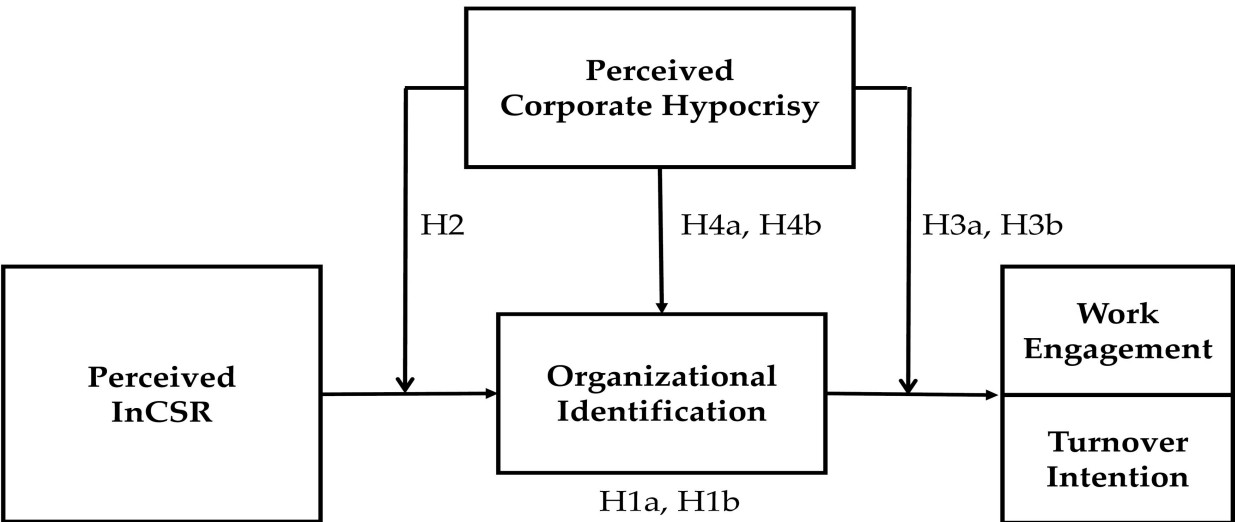

**Figure 1.** Theoretical model.

## 2. Literature Review

### 2.1. Corporate Social Responsibility

The concept of CSR has been widely applied since the 1960s. It emphasizes that the activities that benefit the well-being of society are encouraged and acquiesced to some extent by firms but are beyond the direct interests of firms and shareholders and are not stipulated by laws and regulations [29]. Obviously, the stakeholders are the key factors for the success of CSR activities. Scholars classify CSR activities into internal and external activities targeted at stakeholders inside and outside the firm [30,31]. External CSR (ExCSR) mainly refers to activities focused on local communities, environment, and consumers. Internal CSR (InCSR) refers to a firm's managerial activities focused on internal members [30]. From this viewpoint, employees undoubtedly belong to the most critical role of stakeholders when concerning InCSR practices [32–34]. InCSR practices mainly aim to support the physical and mental wellbeing of employees, such as providing healthy and safe work conditions, protecting employee rights, striking a work–life balance, promoting training and development programs for employees, and creating diverse and equal opportunities according to the development of technologies and talent [35–38]. Research evidence shows that CSR activities carried out by a firm affect employee behaviors and attitude such as job satisfaction [13,39], organizational commitment [11], employee loyalty [15], turnover intention [40,41], organizational identification [42], organizational citizenship behaviors [43,44], and work engagement [45,46].

### 2.2. Employee Perceptions of InCSR

Evidence from literature in the past has shown that ExCSR and InCSR initiatives may have a significant impact on employees' attitudes and behaviors from a micro-level perspective [46–48]. The firm's ExCSR and InCSR practices are important sources from which employees make judgments [49]. ExCSR involves stakeholders outside the organization and has been extensively studied in the literature, but there are few studies from the internal aspect perspective on an employee's perception of benefit related to CSR activities [50]. The trend of InCSR research is increasing in the field of organization [51], and studies involving InCSR are important due to the fact that it often helps attract and retain employees [52]. Previous studies confirmed that CSR policies play a role in motivating employees [53]; however, we focus on the aspect of InCSR for the purpose of our research. Since InCSR focuses on providing services related to employee benefits, it is closely linked to the physical and psychological well-being of employees [2]. Scholars argued that perceived InCSR seems to be self-focused as opposed to ExCSR, due to InCSR practices being able to provide employees with necessary clues that the organization cares

about them [47]. Employees' perception of InCSR emphasize employees' views of their firm's efforts in promoting InCSR practices, rather than evaluating the InCSR plan from the firm's perspective.

It is particularly important for the firm to understand how employees feel, since that feeling exerts a significant impact on their attitude and behaviors at work as well as job performance [54,55]. If employees are not aware of the firm's InCSR practices or cannot sense such actions, they will be indifferent to any other activities of the firm [7,56,57]. To make employees recognize the firm's efforts in implementing InCSR practices, it is necessary to improve employee awareness of InCSR practices, thus making it more likely for employees to meet the firm's expectations [58].

### 2.3. The Mediating Role of Organizational Identification

When the values of employees correspond to those of their organizations, individuals consider themselves an integral part of the organization, thus giving rise to organizational identification [59]. Social identity theory (SIT) is one of the most influential theories in the study of group relation and can be used to explain the correlation between InCSR and organizational identification [60]. Individuals classify themselves and others into various groups based on certain key characteristics such as emotion, age, gender, culture, and other factors [61]. When individuals identify themselves with a certain group, as their sense of belonging gets stronger, their individual self-conception is integrated into their perception of the organization, including its uniqueness, relevance, and persistence [62]. Therefore, they are more willing to act in the best interests of the organization [63]. The main object of InCSR is the employee, whose relationship with the organization is strengthened by the feeling of trust and belonging when he or she senses the organization's concerted efforts in implementing InCSR practices, as they are the most important internal stakeholders. Therefore, they become more supportive of the organization and contribute to its social and economic performance [64].

Although CSR activities carried out by firms are one of the major sources of employee identification, studies on the relationship between CSR and employee-related outcomes rarely examine the psychological mechanism [65]. Organizational identification is an important predictor of work-related outcomes [60,66]. Many studies have pointed out that organizational identification affects employees' attitudes and behaviors at work, such as job satisfaction, task performance, intrinsic motivation, job performance, and organizational citizenship behaviors [26]. However, only a few existing studies discuss the direct relationship between work engagement and organizational identification [59,67,68]. Work engagement is a psychological state wherein employees find meaningfulness, security, and practical value from their work and are, therefore, more willing to fully invest in their work to meet the organization's goals [69]. Similarly, a strong psychological connection between individual and organization can enhance the individual's willingness to perform better and engage more in his or her work [67]. Previous studies have proven that dedicated employees are excited about their job and find meaning in work-related activities and experiences [70], they are more likely to continue with their current employer and have less intention to leave current job [71,72]. Rashid et al. discovered in their studies on the private banking industry that InCSR practices within a firm can negatively impact employees' turnover intention [73]. Moreover, by utilizing questionnaires on professionals in the scientific and technical activities industry in Malaysia, it was also found that InCSR practices within a firm are negatively correlated with employees' turnover intention [40]. Employees with a strong sense of identification with the organization are more likely to stay and try their best to create value for the organization [62,74]. It is on this basis that we propose the following hypotheses:

**Hypothesis 1a (H1a).** *Organizational identification (OI) mediates the positive relationship between employees' perception of InCSR practices and their work engagement.*

**Hypothesis 1b (H1b).** *Organizational identification (OI) mediates the negative relationship between employees' perception of InCSR practices and their turnover intention.*

*2.4. The Moderating Role of Perceived Corporate Hypocrisy*

On the individual psychological level, hypocrisy may be perceived when other people's claims are found to be inconsistent with their actual actions [75]. The concept of hypocrisy, when extended to the managerial field in combination with corporate social responsibility, gives rise to the concept of corporate hypocrisy against the backdrop of CSR. According to Wagner [16], corporate hypocrisy is "the belief that a firm claims to be something that is not." In other words, it implies "saying one thing and doing another". When this perception occurs, it may have a negative impact on consumer and shareholder reactions as well as consequences on the loss of brand assets, sales revenue, and profits [76]. Previous studies mostly focus on exploring and examining how consumers and the market react to corporate hypocrisy [16,77,78]. Consumer perception of corporate hypocrisy often attracts more media attention and may thereafter impact the firm's survival in the business world [18]. However, few studies have explored employees' perception of corporate hypocrisy [17,19]. Obviously, employees are most likely to notice the hypocritical side of their firms, and thus, their organizational identification and work behaviors are likely to be affected by such a perception [19]. Studies suggested that the benefits created by CSR activities on employees' work attitude or behaviors are a result of the mediating effect of organizational identification and trust [64]. If the manager's words and actual deeds are inconsistent, employees will have a distrust of them and reduce their motivation to contribute to the company [79]. This result not only negatively impacts employees but also makes the firm liable to damage. Therefore, when employees perceive their firm to be hypocritical, it may affect their feelings towards the firm and expectation of their work [80], thus reducing their identification toward the firm at the same time [19]. Based on the above, we propose the following hypothesis:

**Hypothesis 2 (H2).** *Perceived corporate hypocrisy moderates the positive relationship between InCSR and OI, such that the higher perceived corporate hypocrisy, the weaker the InCSR–OI relationship.*

As previously mentioned, work engagement is regarded as a psychological state, it is neither a specific mindset that lasts for a short term nor is it about any specific occasion, thing, or individual. It is an emotional and mental state formed over a long period [25]. Given that InCSR practices are voluntary practices conducted by a firm, meaning that the firm provides more than what is required by laws for employees, such as fair job opportunities, a good professional training environment, and career development, thus making employees have a strong feeling of support from their firm [46,81]. Since InCSR sends out positive signals to employees that the organization will look after them and the care from the organization is perfectly just, InCSR initiatives improve employees' self-esteem, optimism, and appreciation, leading to a high degree of work engagement in completing the job tasks assigned by the firm [25].

Scholars believed that work engagement is one of the primary factors leading to a sense of happiness at work [82]. Conversely, voluntary turnover is still a challenge toward organizational management because it may involve significant costs such as loss of professional human resources, increasing expenditure of recruiting and training replacement, and worsening service quality [83–85]. In general, turnover intention occurs prior to voluntary turnover. It is defined as the strength of an employee's intention to quit his or her job [86] and is a cognitive process through which an employee carefully considers, calculates, and plans whether to resign. Previous studies have verified that turnover intention is widely deemed as an important indicator to predict whether an employee will actually leave of

his or her own accord [87,88]. Scholars believe that the implementation of InCSR and the provision of positive working conditions by companies will not only help attract and retain talented employees, but also enhance their identification and commitment with the company [89]. Similarly, InCSR initiatives may increase meaningfulness for employees at work, and that can also reduce their intention to leave the company [90].

When employees are aware of the firm's statement and commitment regarding InCSR practices from various channels, they will have a certain degree of expectation for the firm's behaviors. However, when employees find out that the firm is acting against what it previously claimed, this inconsistency information will cause the employees to re-evaluate whether they share the same values with the firm, or even worse, to think that the firm has a double standard [91,92]. As a result, the employees may feel moral disidentification with the firm, look for ways to disassociate themselves from the firm, or even intend to leave the firm due to emotional stress [92,93]. In light of this, we propose the following hypotheses:

**Hypothesis 3a (H3a).** *Perception of corporate hypocrisy moderates the positive relationship between InCSR and work engagement (WE), such that the higher perception of corporate hypocrisy, the weaker InCSR–WE positive relationship.*

**Hypothesis 3b (H3b).** *Perception of corporate hypocrisy moderates the negative relationship between InCSR and turnover intention (TI), such that the higher perception of corporate hypocrisy, the weaker InCSR–TI negative relationship.*

Based on social identity theory, when individuals identify themselves with the group they are related with, they may have a sense of belonging to the group or identification with the group [59]. If employees see themselves as members of the organization, then they have organizational identification. InCSR provides a foundation for employees to make a comparison of the pros and cons between different groups, enhancing employees' organizational identification [94]. However, when employees find out from various information sources that the firm's words and actual actions are inconsistent, their trust in the positive cues will be queried, while their focus will shift to the negative cues [16]. Scholars have pointed out that when a firm focuses on promoting ExCSR activities for the purpose of turning a quick profit and neglects InCSR, this inconsistent CSR strategy may trigger employees to perceive it as hypocritical, leading to adverse effects, such as emotional burnout and turnover intention [92,93,95]. In addition, the study also found that when employees noticed that the organization's statements were contrary to their actual behavior, the employees would react strongly and negatively [22,96]. Based on the above, we propose the following hypothesis:

**Hypothesis 4 (H4).** *Perceived corporate hypocrisy moderates the indirect effect of perceived InCSR on work engagement (H4a), and turnover intention (H4b) through organizational identification, and the effect will be stronger when perceived corporate hypocrisy is high than when it is low.*

## 3. Materials and Methods

### 3.1. Participants and Design

This study was conducted in a regional teaching hospital in Taiwan. The subjects are medical professional staff employed at the hospital. Why choose a hospital as the research background? The implementation of CSR for hospitals can gain more recognition from their major stakeholders and benefit from operating; it can also improve reputation, increase patient's loyalty, motivate and retain competent employees, and improve overall financial performance [97]. The management of medical institutions in Taiwan is like that of other profit-seeking organizations, these hospitals must be managed to maximize profits [98]. Because the revenue brought by health insurance is rather limited, and the pricing of self-supporting services is restricted by law, it is not an easy task for hospitals to increase income. Most medical institutions strive to cut costs, whether it is a public hospital or a private hospital. In fact, staff is the most important asset of any hospital. If

managers ignore the feelings and expectations of their staff, it will lead to lower morale and higher turnover intention. Moreover, their service quality and customer satisfaction (patients and their families) will be compromised and it may also have a significant impact on the reputation and financial performance of the hospital [99].

In this study, the relationship between variables is shown in Figure 1. Since employees are the major target of this study, the participants were enrolled medical professional staff in hospital, but excluded the supervisor of each unit for the purpose of preventing research bias according to the definition of "employee" by Rupp et al.: "employees as the non-management workforce because these individuals are less likely to be involved in developing and implementing CSR policy" [100]. However, employees can indeed experience the gap between the firm's communication with them and the actual actions of InCSR.

To conduct this survey, we distributed questionnaires by using convenient sampling in each unit of medical department. The questionnaire was placed in a blank envelope without any mark and sent to participants by their supervisor of each unit, then collected in those sealed envelopes by researcher after being answered anonymously. There is no doubt that the identity of the respondent could not be traced back. There were 468 medical professional staff working in the regional teaching hospital. A total of 393 questionnaires were distributed, of which 327 were returned and 311 were analyzed, the effective response rate was 79.13%. All respondents were full-time employees of the hospital, excluding out-sourced personnel. Among them, 88.7% ($n = 276/311$) were females and 11.3% ($n = 35/311$) were males.

### 3.2. Measures

To translate survey instruments, we first selected scales in English from existing research. Next, following the procedure [101], one bilingual author engaged in translation and another performed back-translation for all survey instructions and items, repeating the process until convergence was reached. All the variables (except control variables) were measured on a five-point Likert scale, ranging from "strongly disagree" (1) to "strongly agree" (5).

#### 3.2.1. Employee's Perceptions of InCSR

InCSR to employees was adapted from Turker's instrument [36]. Six items focus on InCSR to employees. A sample item is: "Our hospital policies encourage the employees to develop their skills and careers". Internal consistency reliability was measured as Cronbach's $\alpha = 0.850$.

#### 3.2.2. Organizational Identification

Organizational identification was adapted with a five-item form of an organizational identification scale which was developed by Mael and Ashforth's [68]. A sample item is "When somebody criticizes my working hospital, it feels like a personal insult". Internal consistency reliability was measured as Cronbach's $\alpha = 0.842$.

#### 3.2.3. Perceived Corporate Hypocrisy

Perceived corporate hypocrisy of inconsistency between communication and actual actions was developed and adapted from Wagner [16] and measured with six-items used to gauge corporate hypocrisy. A sample item is "In my opinion, what my working hospital says and does are two different things". Internal consistency reliability was measured as Cronbach's $\alpha = 0.880$.

#### 3.2.4. Work Engagement

To measure employees' work engagement, we relied on the Utrecht Work Engagement Scale (UWES) developed by Schaufeli et al. [102]. The short questionnaire is composed of

9 items that a sample item is "At my work, I feel bursting with energy". Internal consistency reliability was measured as Cronbach's $\alpha$ = 0.713.

### 3.2.5. Turnover Intention

Turnover intentions were measured with a 5-item scale adapted from Wayne et al. [103]. A sample item is "As soon as I can find a better job, I will leave my working hospital". Internal consistency reliability was measured as Cronbach's $\alpha$ = 0.865.

### 3.2.6. Control Variables

To prevent potential confounding effects, this study used age, sex, education, and organizational tenure as control variables. The reason was that previous studies showed these variables may influence outcome variables [12,82,104].

## 4. Results

### 4.1. Reliability and Validity

The measurement model was analyzed using confirmatory factor analysis (CFA), which examined construct reliability and validity. One item (that is "Our hospital encourages its employees to participate to the voluntarily activities") of the InCSR was dropped due to low factor loadings. As shown in Table 1, all the composite reliability (CR) of the constructs ranged from 0.851 to 0.927, exceeding 0.7, indicating that all constructs have internal consistency. Moreover, all average variance extracted (AVE) ranged from 0.525 to 0.578, exceeding 0.5, showing that all constructs have adequate convergent validity.

**Table 1.** Results for the measurement model.

| Construct | Number of Items | CR | AVE |
|:---:|:---:|:---:|:---:|
| Perceived InCSR | 5 | 0.868 | 0.535 |
| Perceived Corporate Hypocrisy | 6 | 0.908 | 0.559 |
| Organizational Identification (OI) | 5 | 0.887 | 0.528 |
| Work Engagement (WE) | 9 | 0.927 | 0.525 |
| Turnover Intention (TI) | 5 | 0.851 | 0.578 |

For the discriminant validity, the square root of the average variance extracted (AVE) was compared. In Table 2, the numbers with round brackets in the diagonal direction represent the square roots of AVEs. Because most of the numbers in the diagonal direction are greater than the off-diagonal numbers, discriminant validity appears to be satisfactory for all constructs.

Model fit indicators determine whether the sample data fit the structural equation model proposed. A variety of standards were recommended [105–108] to determine the model fit of a structural model. The findings in Table 3 show a good model fit in the study: $\chi^2$ = 881.154, $\chi^2$/df = 2.325, RMSEA = 0.065, SRMR = 0.078, TLI = 0.901, CFI = 0.914, GFI = 0.845, AGFI = 0.810.

**Table 2.** Discriminant validity, mean, standard deviation, and correlations for the measurement model.

| Variables | Mean | SD | 1 | 2 | 3 | 4 | 5 | 6 | 7 | 8 | 9 |
|---|---|---|---|---|---|---|---|---|---|---|---|
| 1.Gender [a] | 1.89 | 0.32 | | | | | | | | | |
| 2.Age | 5.03 | 2.17 | −0.005 | | | | | | | | |
| 3.Education | 2.57 | 0.70 | −0.206 *** | −0.153 ** | | | | | | | |
| 4.OT [b] | 9.68 | 8.42 | 0.063 | 0.674 *** | −0.082 | | | | | | |
| 5.InCSR | 3.08 | 0.74 | −0.062 | −0.068 | −0.065 | −0.147 * | (0.731) | | | | |
| 6.CH | 2.95 | 0.70 | 0.055 | −0.004 | 0.038 | 0.071 | −0.759 *** | (0.748) | | | |
| 7.OI | 3.57 | 0.65 | 0.006 | 0.250 *** | −0.091 | 0.203 *** | 0.366 *** | −0.359 *** | (0.727) | | |
| 8.WE | 3.48 | 0.66 | −0.072 | 0.256 *** | −0.032 | 0.088 | 0.405 ** | −0.368 *** | 0.688 *** | (0.725) | |
| 9.TI | 2.80 | 0.87 | 0.097 | −0.164 ** | 0.035 | −0.118 * | −0.413 *** | 0.567 *** | −0.450 *** | −0.422 *** | (0.760) |

Notes: N = 311. * $p < 0.05$ (two-tailed), ** $p < 0.01$ (two-tailed), *** $p < 0.001$(two-tailed). The items on the diagonal with round brackets represent the square roots of the AVE; off-diagonal elements are the correlation estimates. OT: organizational tenure; InCSR: perceived internal CSR; CH: corporate hypocrisy; OI: organizational identification; WE: working engagement; TI: turnover intention; [a] 1: male; 2: female; [b] years of working at current hospital.

**Table 3.** Model fit.

| Model Fit | Criteria | Model Fit of Research Model |
|---|---|---|
| $\chi^2$ | The smaller the better | 881.154 |
| DF | The larger the better | 379 |
| $\chi^2/DF$ | $1 < \chi^2/DF < 3$ | 2.325 |
| RMSEA | <0.08 | 0.065 |
| SRMR | <0.08 | 0.078 |
| TLI | >0.9 | 0.901 |
| CFI | >0.9 | 0.914 |
| GFI | >0.8 | 0.845 |
| AGFI | >0.8 | 0.810 |

Notes: N = 311. $\chi^2$ = chi-square, DF = degrees of freedom, RMSEA = root mean square error of approximation, SRMR = standardized root mean residual, TLI = Tucker–Lewis Index, CFI = comparative fit index; GFI = goodness of fit, AGFI = adjusted goodness of fit.

### 4.2. Descriptive Statistics

Table 4 shows the demographic data for the sample population in this study. Most participants were female (88.7%), mainly because medical staff belonged to the nursing department (N = 189, 60.8%). More than half (59.1%) of the participants were aged below 40 years. The majority (52.4%) of the respondents had a bachelor's degree, followed by an associate's or under (42.1%). Most of the job tenure at the current hospital was 1–5 years (27.97%), followed by 5–10 years (18.97%). There were also 17.69% of medical staff who had worked in this hospital for more than 20 years.

The means, standard deviations, and correlations between the different variables in this study are shown in Table 2. From the table, there is no significant correlation between gender and various research variables, while age and organizational tenure in the hospital are significantly positively correlated with OI. Furthermore, InCSR was significantly positively correlated with OI, but negatively correlated with corporate hypocrisy and TI. However, perceived corporate hypocrisy negatively correlated with OI and WE, but significantly positively with TI.

**Table 4.** Descriptive statistics of sample (N = 311).

| Description | Frequency | Percentage (%) |
|---|---|---|
| Gender | | |
| Male | 35 | 11.3 |
| Female | 276 | 88.7 |
| Age | | |
| 30 or under | 95 | 30.5 |
| 31–40 | 89 | 28.6 |
| 41–50 | 76 | 24.5 |
| 51–60 | 46 | 14.8 |
| 61 or above | 5 | 1.6 |
| Education | | |
| Associate or under | 131 | 42.1 |
| Bachelor | 163 | 52.4 |
| Master or above | 17 | 5.5 |
| Organizational tenure | | |
| less than 1 years | 107 | 12.54 |
| 1–5 years | 87 | 27.97 |
| 5–10 years | 59 | 18.97 |
| 10–15 years | 42 | 13.50 |
| 15–20 years | 29 | 9.33 |
| >20 years | 55 | 17.69 |

### 4.3. Mediation Analysis Results

In this study, the mediation effect was tested using PROCESS macro for SPSS (version 3.4) (model 4) developed by Hayes, A.F. [109], 5000 bootstrap samples, and 95% confidence interval (CI) as recommended [110]. The results are shown in Table 5. The total effect of InCSR on WE had a significant positive relationship ($\beta$ = 0.376, $p < 0.001$), the indirect effect was significant ($\beta$ = 0.216, 95% CI: [0.145, 0.291]), and the direct effect was also significant ($\beta$ = 0.161, $p < 0.001$). OI had a mediation effect between InCSR and WE. On the other hand, the total effect of InCSR on TI had a significant negative relationship ($\beta$ = −0.510, $p < 0.001$), the indirect effect was significant ($\beta$ = −0.146, 95% CI: [−0.233, −0.073]), and the direct effect was also significant ($\beta$ = −0.365, $p < 0.001$). OI also has a mediation relationship between InCSR and TI. Hence, hypothesis 1a and 1b were supported.

**Table 5.** Results for mediation effect of organizational identification.

| Independent Variable | Mediator | Effect | Dependent Variable: WE | | | | Dependent Variable: TI | | | |
|---|---|---|---|---|---|---|---|---|---|---|
| | | | $\beta$ | SE | t | 95% CI | $\beta$ | SE | t | 95% CI |
| InCSR | OI | Total effect | 0.376 *** | 0.045 | 8.288 | [0.287, 0.466] | −0.510 *** | 0.061 | −8.385 | [−0.630, −0.391] |
| | | Direct effect | 0.161 *** | 0.039 | 4.088 | [0.083, 0.288] | −0.365 *** | 0.063 | −5.748 | [−0.489, −0.240] |
| | | Indirect effect | 0.216 | 0.037 | | [0.145, 0.291] | −0.146 | 0.040 | | [−0.233, −0.073] |

* $p < 0.05$, ** $p < 0.01$, *** $p < 0.001$.

### 4.4. Moderation and Moderated Mediation Analysis Results

This moderation and moderated mediation effect were also tested using PROCESS macro for SPSS (version 3.4) and 5000 bootstrap samples with 95% confidence interval.

Table 6 provides the results of whether perceived corporate hypocrisy had a moderating effect between InCSR and OI, WE, and TI. The first independent variable of the test was OI. The coefficient of determination ($R^2$) is 28.6%, and the regression coefficient of perceived InCSR $\times$PCH = −0.197, $p < 0.001$, indicating that perceived corporate hypocrisy has a significant negative moderating effect between the InCSR and OI. When the perception of corporate hypocrisy is higher, the positive influence of InCSR on OI will be weakened (Figure 2A), hence hypothesis 2 was supported.

**Table 6.** Moderating effect of perceived corporate hypocrisy.

| Independent Variable | Dependent Variable | | | | | |
|---|---|---|---|---|---|---|
| | Organizational Identification | | Work Engagement | | Turnover Intention | |
| | B (SE) | 95%CI | B (SE) | 95%CI | B (SE) | 95%CI |
| InCSR | 0.864 (0.144) *** | [0.580, 1.148] | 1.194 (0.138) *** | [0.922, 1.467] | −0.565 (0.179) ** | [−0.917, −0.212] |
| Perceived corporate hypocrisy (PCH) | 0.455 (0.143) ** | [0.174, 0.736] | 0.769 (0.137) *** | [0.499, 1.038] | 0.155 (0.177) | [−0.193, 0.504] |
| InCSR ×PCH | −0.197 (0.041) *** | [−0.278, −0.117] | −0.285 (0.039) *** | [−0.362, −0.207] | 0.184 (0.051) *** | [0.084, 0.284] |
| R² | 0.286 | | 0.366 | | 0.384 | |
| F | 17.374 *** | | 24.989 *** | | 26.956 *** | |

* $p < 0.05$, ** $p < 0.01$, *** $p < 0.001$.

Similarly, when the independent variable was WE, the coefficient of determination ($R^2$) was 36.6%, and the regression coefficient of InCSR ×PCH = −0.285, $p < 0.001$, indicating that perceived corporate hypocrisy had a significant negative moderating effect between the InCSR and WE. When the perception of corporate hypocrisy was higher, the positive influence of InCSR on WE was weakened (Figure 2B), hence hypothesis 3a was supported. When the dependent variable was TI, the coefficient of determination ($R^2$) was 38.4%, and the regression coefficient of InCSR ×PCH = 0.184, $p < 0.001$, indicating that perceived corporate hypocrisy had a significant positive moderating effect between the InCSR and TI. When the perception of corporate hypocrisy was higher, the negative impact of InCSR on TI was weakened (Figure 2C). Namely, the higher the InCSR, their low TI will not be reduced, so hypothesis 3b was supported.

This study explores the effect of moderate mediation from perceived corporate hypocrisy to OI, using bootstrapping through PROCESS macro for SPSS, version 3.4 (model 8). It can be seen from Table 7 that the indirect effect of InCSR and corporate hypocrisy on WE through the mediated effect of OI was −0.108 (SE = 0.041, 95%CI [−0.197, −0.039]), not across 0; the perceived corporate hypocrisy had a significant weakening on the mediating effect of OI. That is, when the perception of corporate hypocrisy was high, the mediation effect of OI between InCSR and WE was weaker than when perceived corporate hypocrisy was low. In other words, the impact of InCSR on WE through OI was affected by the perception of corporate hypocrisy. Therefore, hypothesis 4a was supported. On the other hand, the indirect effect of InCSR and corporate hypocrisy on TI through the mediated effect of OI is 0.061 (SE = 0.025, 95%CI [0.019, 0.115]), not across 0, the perceived corporate hypocrisy had a significant weakening on the mediating effect of OI. That is, when the perception of corporate hypocrisy was high, the negative mediation effect of OI between InCSR and TI was weaker than when perceived corporate hypocrisy was low. In other words, the impact of InCSR on TI through OI was affected by the perception of corporate hypocrisy. Therefore, hypothesis 4b was supported.

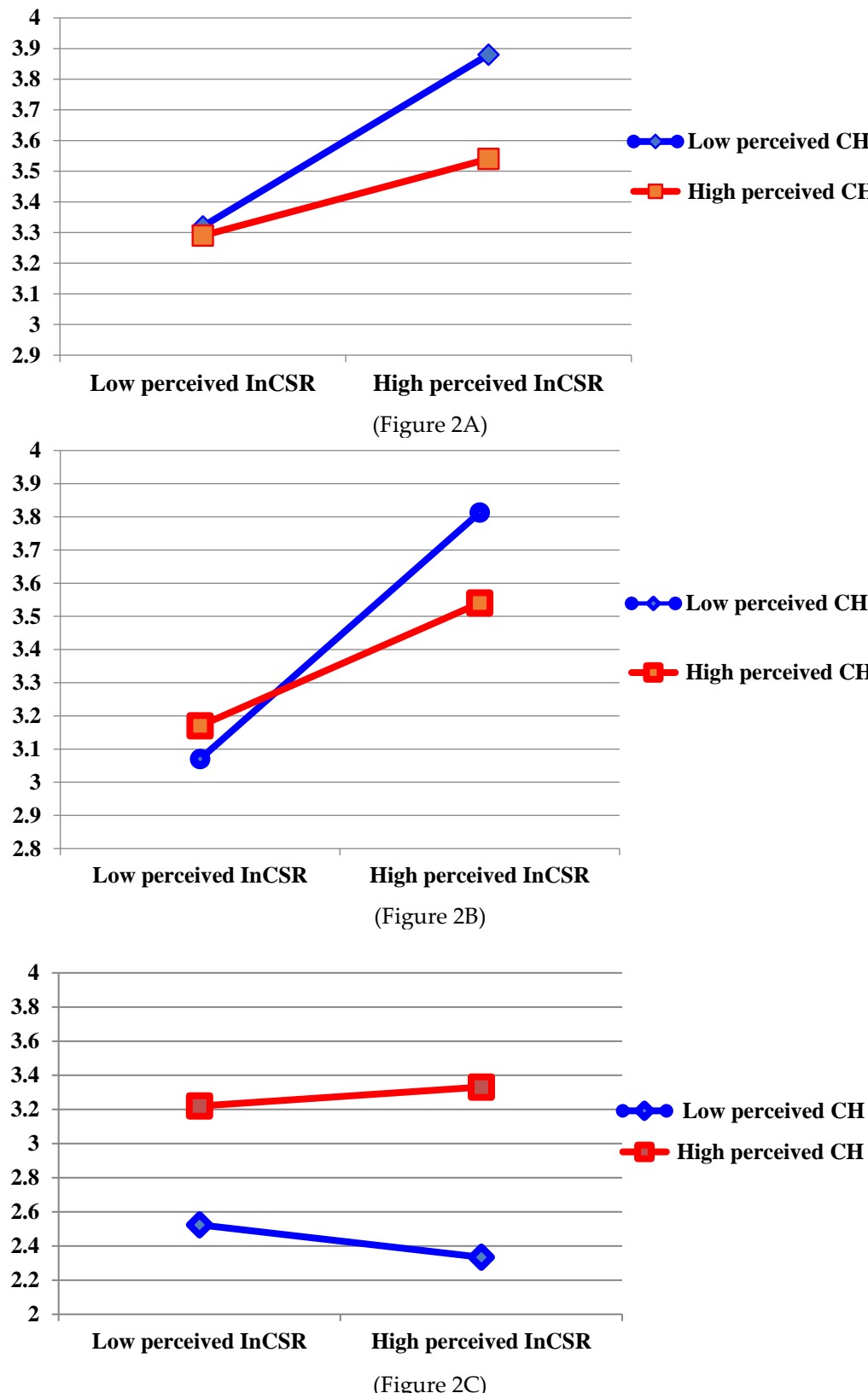

**Figure 2.** (**A**) Interaction of perceived corporate hypocrisy (CH) and perceived InCSR on organizational identification (OI); (**B**) interaction of perceived corporate hypocrisy (CH) and perceived InCSR on work engagement (WE); (**C**) interaction of perceived corporate hypocrisy (CH) and perceived InCSR on turnover intention (TI).

**Table 7.** Moderated mediation effect of perceived corporate hypocrisy on organizational identification.

| Independent Variable | Dependent Variable | | | | | |
| --- | --- | --- | --- | --- | --- | --- |
| | OI | | WE | | TI | |
| | b (SE) | 95%CI | b (SE) | b (SE) | 95%CI | b (SE) |
| Total effect | | | | | | |
| InCSR | 0.864 (0.144) *** | [0.580, 1.148] | 0.722 (0.120) *** | [0.485, 0.959] | −0.298 (0.184) | [−0.660, 0.063] |
| PCH | 0.455 (0.143) ** | [0.174, 0.736] | 0.520 (0.115) *** | [−0.048, 0.639] | 0.296 (0.175) | [−0.048, 0.639] |
| OI | | | 0.547 (0.045) *** | [−0.444, −0.172] | −0.308 (0.069) *** | [−0.444, −0.172] |
| Indirect effect | | | | | | |
| InCSR ×PCH | −0.197 (0.041) *** | [−0.278, −0.117] | −0.177 (0.033) *** | [−0.242, −0.111] | 0.123 (0.051) * | [0.023, 0.223] |
| Moderated mediation effect of PCH | | | −0.108 (0.041) | [−0.197, −0.039] | 0.061 (0.025) | [0.019, 0.115] |
| $R^2$ | 0.286 | | 0.572 | | 0.422 | |
| F | 17.374 *** | | 50.436 *** | | 27.544 *** | |

* $p < 0.05$, ** $p < 0.01$, *** $p < 0.001$.

## 5. Discussion

This study used perceived corporate hypocrisy as the moderating variable and organizational identification as the mediator to examine the impacts of words and actions inconsistency about the perception of InCSR initiatives by employees on their work engagement and turnover intention. Social identity theory was the main basis of discussion. It also examined whether perceived corporate hypocrisy moderates the mediating effect of organizational identification. The research results provide more insight than just the direct effects of InCSR practices on the outcomes of employee behaviors, and the findings verify the important effect between the mediator and moderating variable.

Firstly, based on the research findings, we verify that employees' work engagement will be strengthened when there is a higher degree of perception of InCSR practices. This is in line with the conclusion of previous studies; that is, when a firm implements its plan for InCSR, employees will be attracted by such actions, thus increasing employees' willingness to identify with the firm, work harder, and demonstrate a proactive work attitude [25,65,111]. In addition, our research also showed that the perception of InCSR practices negatively affects employees' turnover intention. This negative correlation is also consistent with findings of previous studies [73,92,112]. As pointed out by Rupp et al. [113], employees' perception of CSR practices in the firm will affect their work engagement. Moreover, the degree of employee engagement is determined by the firm's follow through on its CSR statement and commitment, as well as its involvement in related activities. For employees, perceived InCSR is more direct and carries more symbolic meaning than real actions. When employees are more engaged at work, they are more willing to work hard for their organization, meanwhile, their productivity and performance will improve and they will be less likely to leave their current organization [114–116].

Secondly, this study found that organizational identification has a significant and positive mediating effect on the relationship between employees' perception of InCSR practices and their work engagement. At the same time, it exerts a significant and negative mediating effect on the relationship between employees' perception of InCSR practices and their turnover intention. How does InCSR affect employees' behavior and attitudes at work? Most scholars utilize social identity theory or social exchange theory to explain the mediating relationship, such as organizational trust [117], organizational justice [12,118], perceived organizational support [46,119], organizational identification [25,120], and organizational commitment [40,112]. This study confirms that organizational identification

has a significant mediating effect on the relationship between employees' perception of InCSR practices and their work engagement, as well as turnover intention. Since the main targets of InCSR initiatives are employees, they will be able to feel that the firm respects them. Such a feeling is unique and makes the firm more appealing as compared to other organizations. Moreover, employees will therefore identify with the firm, thus enhancing their sense of organizational identification [2,47]. An employee with a relatively strong sense of organizational identification will be more willing to dedicate more to work [27]. Previous studies show that engaged employees will be excited about work, find meaning in work-related activities and experiences [70], and hence be more willing to keep working for their current employer [72] and less likely to leave [71].

Although this study confirms that the perception of InCSR practices does enhance employees' organizational identification, and therefore generate positive outcomes with respect to work behaviors, while lowering the turnover intention, this study also explored what the consequences would be if employees find out that the firm is not following through on its claims and perceive this inconsistency as hypocritical. Table 7 and Figure 2 revealed that the InCSR practices taken by the firm indeed positively affect their organizational identification with a low degree of employee's perceived corporate hypocrisy. However, the positive effect of organizational identification then significantly weakened with a high degree of perceived corporate hypocrisy. This phenomenon indicates that the InCSR practices of a firm aimed at improving the overall well-being and future development for its employees may send a message or a cue to the employees, namely that their firm cares about and values them [2]. In addition, employees' self-awareness will be integrated into the firm's values and policies, resulting in a high degree of organizational identification [121]. Organizational identification fundamentally changes the relationship between employee and organization and affects their work performance [122]. Nonetheless, employees will continuously evaluate whether the organization keeps to its word that was stated internally. When employees notice that the organization is acting contrary to what it previously expressed, they tend to focus more on the negative message. Moreover, if they judge these actions inconsistently or feel betrayed, then they might respond negatively [16, 49]. The research findings verify that, when there is a higher degree of perception of the firm as hypocritical among employees, the positive mediating effect generated by organizational identification will be affected and weakened, then their work engagement will be compromised, and turnover intention will be strengthened.

## 6. Conclusions and Limitations

### 6.1. Theoretical Implications

This study extended the boundary of literature on CSR at the micro-level and provides important contributions. Since InCSR practices are those actions taken by a firm with regards to its employees, previous literature described strategies concerning InCSR practices, including employee trainings, continuing education plans, safe work environments, day care programs, and ethics guides for employees' activities [2,11,36]. However, past studies focus on the impacts of InCSR practices on subsequent employee-related results and found a number of mediators in the process. Our research findings show that InCSR practices indeed positively affect employees' work attitudes, including increasing their work engagement and reducing turnover intention. This study also confirms that organizational identification does have a mediating effect on the relationship between the perception of InCSR practices and outcomes related to employees.

The second contribution of this study to the literature is providing theoretical support for the research on the "dark side" of corporate social responsibility at the individual level of analysis. Most of the previous studies on corporate hypocrisy focus on the reaction of consumers and impacts on the firms [22,77,123]. This study extends the focus to internal stakeholders (employees) of firms and the work-related influence on their behavior and attitudes. Research findings reveal that, when employees perceive the firm to be hypocritical

due to inconsistency between words and actions, their work engagement will be negatively impacted, while increasing their turnover intention.

In addition, previous studies have pointed out that investing in InCSR practices will help to obtain positive feedback related to employees' identification, in particular, identification with the organization [9,64]. However, the evolving process of identification often involves the transformation of values and beliefs [124]. Therefore, the gap will be developed when the actual perception fails to meet employees' expectations and inconsistent feeling will emerge [125]. This study utilizes the perception of corporate hypocrisy as the moderating variable to explore the degree of impact on the mediator, that is, organizational identification, which is supposed to create a significant effect. The findings verify that employees' perception of corporate hypocrisy indeed reduces the mediating effect of organizational identification, negatively impacting their subsequent job performance and intention to stay with the firm.

### 6.2. Practical Implications

Understanding the needs of employees is extremely important for CSR strategy creation. Whereas well-designed CSR initiatives can satisfy the multitudes of relational, heuristic, and deontic needs in employees [57], our study provides insights into the investment in InCSR practices and how an organization should weigh the pros and cons. The findings also make important contributions to practical matters. First, work engagement is a positive emotional state and the antithesis of work burnout [126]. Scholars advocate for firms to promote work engagement instead of preventing burnout. Because it is not enough to merely prevent burnout, firms need to further encourage employees to become more engaged in their work [127]. Therefore, engaged employees tend to reduce their willingness to leave employment and are more likely to continue with their current employer [71,72]. Since the costs associated with quitting and recruiting new employees will produce substantial harm to the organization [88], our research results provide empirical support; that is, when employees perceive the InCSR practices and feel that the firm values them, they will be more engaged in working and tend to reduce the potential influence on turnover intention.

Secondly, firms need to address the concerns of internal and external stakeholders apart from profits; therefore, most firms pay attention to CSR activities. However, when firms carry out internal and external CSR dimensions, what are the factors that may affect employees' attitude and behaviors at work? Ferreira and Oliveira [128] conducted a survey to examine the relationship between CSR and work engagement. The results of their research showed that employees who are exposed to InCSR are more engaged than those who are exposed only to external CSR practices. Findings of our study highlight the intrinsic mechanisms between InCSR and the attitudes and behaviors of employees. According to the empirical evidence provided by studies on marketing and organizational behaviors, CSR can create value and improve organizational performance by establishing a stronger relationship between firms and stakeholders [9,129]. However, for employees, perceived InCSR practices may lead them to enhance valuation about their job and generate positive returns for the organization [25]. This study verifies that there is indeed a considerable correlation between InCSR practices and work attitudes. Given that our research targets are InCSR practices (such as voluntarily activities, skills and careers development, work–life balance, fair management decision, additional education support, etc.), the findings will help managers develop and formulate effective strategies for InCSR practices.

Finally, the findings of our study suggest that when employees perceive the firm as hypocritical, this perception may actually negatively impact their attitude and behaviors at work. The negative impact can be direct or exert influence via the mediating effect of organizational identification. When employees feel that they are in a hypocritical environment, they may be upset and eventually quit their job or lose work engagement, which means they no longer care about the organization [28]. Our results can send a message to managers and decision makers, even though employees are motivated at

their job by the connection to organization through an internal CSR portfolio; InCSR practices that are inconsistent with the firm's previous statement can cause employees to perceive corporate hypocrisy. Such inconsistent perceptions among employees may damage the organization's CSR strategy, and discontent employees may negatively impact other stakeholders, e.g., customers, which may have detrimental consequences for the company. This viewpoint is in line with Scheidler et al.'s study on InCSR [130].

*6.3. Research Limitations*

Although we obtained a few results from this research, there are still some limitations and suggestions for the direction of future explorations. First, the database of this study was obtained from a public hospital, which cannot completely represent all kinds of hospitals, such as private hospitals or hospitals at different levels. Moreover, since we mainly focus on the healthcare industry, the research findings may not be able to represent other industries.

Second, due to the impact of cultural factors and different healthcare systems, various countries have different healthcare policies and work cultures; therefore, the same research design may produce different results.

Third, our study was designed from a cross-sectional perspective and data obtained from one point in time; there may be a problem of common method variance, and it is difficult to accurately estimate the causal relationship between the independent variable and the dependent variables with the data collected at the same time. For future studies, it is recommended that longitudinal or empirical research be carried out to examine the causal relationship between variables to provide clearer conclusions.

In conclusion, CSR can be said to be a commitment of enterprises to contribute to sustainable development in order to improve the quality of life of stakeholders. The concern topics of CSR are mainly focused on economy, environment, and society. However, from the perspective of sustainable development, to satisfy shareholders, customers, collaborations, or the community, everything must be done through employees. Therefore, caring for employees and improving their welfare and happiness may be the most important lesson for companies to implement their internal corporate social responsibilities.

**Author Contributions:** Conceptualization, Y.-P.C. and H.-H.H.; formal analysis, Y.-P.C. and H.-H.H.; investigation, Y.-P.C. and H.-H.H.; methodology, Y.-P.C. and H.-H.H.; validation, Y.-P.C. and H.-H.H.; writing—original draft, Y.-P.C., H.-H.H. and C.-M.L.; writing—review and editing, Y.-P.C. and H.-H.H. All authors have read and agreed to the published version of the manuscript.

**Funding:** This study did not receive any research funding.

**Institutional Review Board Statement:** The study protocol used was reviewed and approved by the Institutional Review Board of Taipei Veterans General Hospital (No. 2018-07-033AC).

**Informed Consent Statement:** Not applicable.

**Data Availability Statement:** The datasets generated and analyzed during the current study are not publicly available due to privacy/ethical restrictions but are available from the corresponding author on reasonable request.

**Conflicts of Interest:** The authors declare no conflict of interest.

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
