# Peer review of "Consistency or Hypocrisy? The Impact of Internal Corporate Social Responsibility on Employee Behavior: A Moderated Mediation Model"

_sustainability, doi:10.3390/su13179494_

Round 1

Reviewer 1 Report

I would like to see the reliability and validity scores of the questionnaires from original researchers who constructed the measurement tools. Also sample size calculation is missing. I know that 311 is a good number, but still a calculation mentioning the population and calculated sample size would make the research more robust.

Reviewer 2 Report

Dear authors, the research you have presented is very interesting and pertinent. Theoretically is well-grounded. The methodology is adequate. The sample is representative. Obtained results are well discussed. Theoretical implications and practical implications are provided, as well as research limitations and directions for future research. Overall, a very good paper. In my opinion, you can improve your paper, so, pay attention to the following:

-Provide the structure of the paper at the end of the Introduction section.

-Citation within the text is not in accordance with the journal's requirements (square brackets and in order of appearing in the text, 1, 2, 3... and so on).

-Re-check manuscript for typos, for example, line 635 has diff. pt., 606-620, and so on.

Improve the technical aspect of the paper and pay attention to the details.
